# Inter-limb asymmetries are associated with decrements in physical performance in youth elite team sports athletes

Azahara Fort-Vanmeerhaeghe[1,2,3]*, Chris Bishop[4], Bernat Buscà[1‡], Joan Aguilera-Castells[1‡], Jordi Vicens-Bordas[5‡], Oliver Gonzalo-Skok[6]

**1** Faculty of Psychology, Education Sciences and Sport (FPCEE) Blanquerna, Ramon Llull University, Barcelona, Spain, **2** School of Health Sciences (FCS) Blanquerna, Ramon Llull University, Barcelona, Spain, **3** Segle XXI Female Basketball Team, Catalan Federation of Basketball, Esplugues de Llobregat, Spain, **4** London Sport Institute, Middlesex University, London, United Kingdom, **5** School of Health and Sport Sciences (EUSES), Universitat de Girona, Salt, Spain, **6** Faculty of Health Sciences, University of San Jorge, Zaragoza, Spain

☯ These authors contributed equally to this work.
‡ These authors also contributed equally to this work.
* azaharafv@blanquerna.url.edu

## Abstract

Actually, there is scarce literature looking for the relationship between inter-limb asymmetries and performance in youth elite team sports. The main purpose of this cross-sectional study was to examine the relationships between inter-limb asymmetries and physical performance in youth elite team-sports players. A secondary objective was to evaluate the presence of between-sexes differences in inter-limb asymmetries in elite youth team sports players. Eighty-one young elite team-sports athletes (age: u-14 to u-18) performed the star excursion balance test in the anterior direction (SEBT ANT), a single leg vertical countermovement jump test (SLCMJ), the one leg hop test for distance (OLHT), a 30 m sprint test, and the V-cut test. Inter-limb asymmetries were calculated for SEBT ANT, SLCMJ and OLHT. Pearson $r$ was used to analyse the relationships between inter-limb asymmetries and physical performance. Results showed significant ($p < 0.05$) but small ($r = 0.26$) relationships between SLCMJ asymmetries and 30 m sprint time for the total group. Significant negative correlations with small to moderate magnitude of correlation were also found between SLCMJ asymmetries and SLCMJ performance on the lowest performing limb for the total group ($p < 0.05$; $r = -0.26$), males ($p < 0.01$; $r = -0.48$) and females ($p < 0.05$; $r = -0.30$). Moreover, significant negative correlations with moderate and large magnitude were also present between OLHT asymmetries and OLHT performance on the lowest performing limb for the total group ($p < 0.01$; $r = -0.44$), males ($p < 0.01$; $r = -0.56$) and females ($p < 0.01$; $r = -0.64$). No correlations were observed between asymmetries and either the V-cut test or SEBT ANT performance. No correlation were observed between SEBT ANT asymmetries and physical performance. In addition, when comparing asymmetry values between sexes there were no significant differences in vertical (p = 0.06) and horizontal (p = 0.61) jumping tests. However, there were significant differences in asymmetry between sexes in the ANT SEBT ($p = 0.04$). In conclusion, the current study indicated that jumping asymmetries were

**Data Availability Statement:** All files are available from the Figshare database: http://dx.doi.org/10.6084/m9.figshare.8282852

**Funding:** The author(s) received no specific funding for this work.

associated with decrements in sprint speed and jumping performance. Therefore, assessing inter-limb asymmetries would be recommended to improve training interventions for youth elite team-sports athletes.

## Introduction

Team-sports are characterized by high intensity unilateral actions such as jumping and changes of direction [1]. Such actions, in addition to limb dominance (e.g., preferred kicking limb) are likely to result in inter-limb asymmetries developing [2–4]. Between-limb differences in power and strength have been purported as important risk factors for sport injuries [5,6] and, in some instances, have been associated to decrements in sporting performance [3,7]. In any given task, the reduced physical capacity of the weaker limb to both produce and absorb force is likely to increase the risk of injury. This is because it is likely to exceed its "tolerance capacity" sooner than the stronger limb when repeated high intensity actions are considered [5,8]. Equally, Maloney [9] has highlighted that practitioners should consider the weaker limb as having a greater "window of opportunity" to increase its capacity. In doing so, minimizing asymmetry may merely be a consequence of targeted training interventions which aim to address deficits in the weaker limb. With asymmetries evident across a range of team sports such as basketball [10–12], soccer [13] and volleyball [14,15], it is surprising that there are currently no clear conclusions about how inter-limb asymmetries affect athletic performance [9].

Recently, there has been a rise in the number of studies investigating the association between inter-limb asymmetry and measures of athletic performance. For example, Lockie et al. [16] reported between-limb differences for jump height during the single leg counter-movement jump (SLCMJ– 10.4%), and distance during the single leg broad jump (SLBJ– 3.3%) and single leg lateral jump (SLLJ– 5.1%) tests in male collegiate athletes. No significant correlations were reported between asymmetry and linear speed or change of direction (COD) speed. Similarly, Dos'Santos et al. [17] reported mean inter-limb differences of 5–6% for jump distance during the single and triple hop tests in male collegiate athletes, and also showed no significant relationships with total time during two COD tests. In contrast, Bishop et al. [7] showed that jump height asymmetry (12.5%) from the SLCMJ was associated with slower 5m ($r = 0.49$; $p < 0.05$), 10m ($r = 0.52$; $p < 0.05$) and 20m ($r = 0.59$; $p < 0.01$) sprint performance in youth female soccer players. In addition, Bishop et al. [18] showed that drop jump asymmetries were associated with slower acceleration, speed and COD performance in adult female soccer players. Specifically, jump height asymmetry was correlated with 30m ($r = 0.58$; $p < 0.05$) and 505 ($r = 0.52–0.66$; $p < 0.05$), and reactive strength index asymmetry was correlated with 10m ($r = 0.52$; $p < 0.05$) and 505 on both limbs ($r = 0.54–0.55$; $p < 0.05$). Thus, with conflicting literature surrounding the association between asymmetry and athletic performance, further research in this area is warranted.

Further to the conflicting evidence relating to asymmetry and physical performance, the majority of literature on this topic has also been conducted in adult populations. Atkins et al. [19] reported peak force asymmetry between 4–13% during a bodyweight overhead squat in elite academy soccer players. Read et al. [20] concluded that the stage of maturation showed no significant differences on between-limb asymmetries during the SLCMJ, SLBJ, 75% hop and Y-balance test, also in youth male soccer players. From this limited evidence, it is clear that male youth populations have been tested; however, minimal literature exists investigating how asymmetries differ between sexes. To the authors' knowledge, only one study has provided such a comparison and this was done in adult collegiate athletes. Bailey et al. [21]

observed that females produced significantly greater inter-limb asymmetry compared to males during loaded and unloaded bilateral jumping tasks. Specifically, females produced larger imbalances during peak force (3.78 vs. 1.95%) and peak power (4.18 vs. 1.81%) during the squat jump, and during peak force (6.89 vs. 4.65%), peak power (15.87 vs. 8.48%) and net impulse (13.50 vs. 6.62%) during the CMJ. Despite the usefulness of this evidence, minimal literature exists pertaining to asymmetry in youth athlete populations and how these may differ between sex.

Therefore, the main objective of this study was to examine the relationship between inter-limb asymmetries and physical performance (Dynamic balance, jumping, linear sprinting, and change of direction speed) in elite youth team-sports players. A secondary objective was to evaluate the presence of between-sexes differences in inter-limb asymmetries in elite youth team sports players. A true hypothesis was challenging to generate given the conflicting and limited literature [21–23]; however, it was hypothesized that youth female athletes would show larger between-limb asymmetries and larger imbalances would be associated with reduced physical performance.

## Methods

### Design

The current study employed a cross-sectional design to examine the relationship between inter-limb asymmetries with physical performance in a group of youth elite team-sports players. Dynamic balance, vertical and horizontal unilateral jumping, linear sprinting, and change of direction speed were assessed with the anterior star excursion balance test (SEBT ANT), SLCMJ, one leg hop test (OLHT), 30 m sprint, and V-cut test, respectively. All tests were performed in September 2017. In addition, the assessment of inter-limb asymmetries were calculated through unilateral tests (i.e., dynamic balance and vertical and horizontal jumping).

### Participants

Eighty-one competitive team-sports players, specifically basketball (22 females), volleyball (14 females and 15 males) and handball (15 females and 15 males), volunteered to participate in this study with athletes tested in the pre-season period. Subjects were eligible for inclusion if they were elite team-sports players between 14–18 years old. Subjects were excluded if they presented any injury (overuse or acute) at the time of testing. Table 1 provides subject characteristics. Routine training did not differ between groups and consisted of 8–10 sessions per week with 90–120 minutes per session, plus a weekend match, totaling approximately 16–20 hours of combined training and competition per week. All the participants train and study in the same high performance sports center, in Esplugues de Llobregat (Joaquim Blume

**Table 1. Participants characteristics as total and split by sex.**

|  | Total ($n$ = 81) | Males ($n$ = 30) | Females ($n$ = 51) |
|---|---|---|---|
| Age (years) | 15.9 ± 1.11 | 16.1 ± 1.07 | 15.8 ± 1.13 |
| Years post-PHV* | 2.72 ± 1.77 | 1.56 ± 1.20 | 3.43 ± 0.94 |
| Body mass (kg) | 69.7 ± 11.6 | 75.4 ± 13.8 | 66.4 ± 9.21 |
| Height (m) | 1.78 ± 0.19 | 1.80 ± 0.32 | 1.77 ± 0.08 |
| BMI (kg·m$^{-2}$) | 20.3 ± 5.14 | 21.1 ± 5.03 | 19.7 ± 5.27 |
| Training experience (years) | 6.13 ± 3.00 | 4.73 ± 3.47 | 6.83 ± 2.47 |

* Estimation of biological age (48)

Residence). Biological maturation was calculated in a noninvasive manner using a regression equation comprising measures of age, body mass, standing height, and sitting height [24]. Prior to the commencement of the study, subjects and their parents received detailed written and verbal information about the possible risks and benefits associated with testing. Written informed consent and assent were obtained from both parents/tutors and participants, respectively. The Catalan Sport Council Ethics Committee approved the study (07/2017/CEICEGC), and it conformed to the recommendations of the Declaration of Helsinki.

## Procedures

One week before data collection, all participants were familiarized with performance tests procedures enabling them to practice each test between 2–5 times. During testing days, all participants completed the same 10-minute neuromuscular warm-up consisting of the following exercises: multidirectional movements combined with strength and dynamic stretching exercises and maximal and progressive intensity displacements including changes of direction, jumps, and acceleration/deceleration movements. Following the warm-up, subjects were allowed two practice trials for each test. Consistent feedback was provided throughout to ensure proper technique. Testing was performed in two separate days with 24–48 hours in-between. In the first day were conducted SEBT, SLCMJ and OLHT while in the second day 30-m sprint and V-cut test.

## Star excursion balance test for the anterior reach (SEBT ANT)

The SEBT ANT measurement was used to asses dynamic balance [25], which is defined as the capacity to maintain center of mass over a fixed base of support under a challenge; specifically, motion of other limbs and body segments, or unanticipated disturbance to supporting surface [26]. Anterior reach was only included because of the higher associations with injury of this direction in youth athletes [27]. This test was performed with socks on the feet and only in the anterior direction [25]. The distal aspect of the subject's big toe was centered at the junction of the grid. While maintaining a single-leg stance, each subject was asked to reach lightly with the contralateral leg in the anterior (ANT) direction. Hands were placed on hips and both controlled stance foot and trunk movement were allowed. The maximal reach distance was measured at the point where the most distal part of the foot touched the line. The trial was dismissed and repeated if the subject failed to maintain unilateral stance, lifted or moved the stance foot from the grid, touched down with the reach foot, or failed to return the reach foot to the starting position. Minimal stance foot movement in the form of slight raising of the forefoot and/or heel from the surface was allowed [25]. The mean reach was used for analysis. For the three trials, participants started with their preferred leg and the order of the right and left legs was alternated thereafter. Each trial was separated by a 30 s recovery period.

## Single leg countermovement jump (SLCMJ)

Subjects were instructed to stand on one leg, descend into a countermovement, and then rapidly extend the stance leg to jump as high as possible in the vertical direction [28]. The swing of the opposite leg prior to the jump was not allowed. They were also instructed to land on both feet simultaneously. A trial was considered successful if the hands remained on the hips throughout the movement and if balance was maintained for at least three seconds after landing. The SLCMJ distance in meters was calculated from flight time [29] with a contact mat system (Chronojump Boscosystem, Barcelona, Spain) [30]. For the three trials of each jump, participants started with their preferred leg and the order of the right and left legs was

alternated thereafter. Each trial was separated by a 30 s recovery period. The best trial of the three jumps was used to statistical analysis.

### One leg hop for distance (OLHT)

The one leg hop for distance is used to estimate unilateral jump capacity. This test has been found to be reproducible and valid [31,32]. All participants were trainers and were asked to hop as far as possible, taking off and landing on the same foot and keeping their balance on this foot for 2 seconds after landing. To facilitate body balance, participants performed the OLHT with free arms. For the two trials of each jump, participants started with their preferred leg and the order of the right and left legs was alternated thereafter. If participants performed 2 invalid jumps with one leg, they had the opportunity to jump as many times as necessary until they kept their balance. Each trial was separated by a 30 s recovery period. The greatest distance for each leg was recorded.

### 30-m sprint

Maximum sprint speed was assessed by 30-m sprint. The start and finish lines were clearly marked with cones. The front foot was placed 0.5 m before the first timing gate. Each player completed two sprints with a three-minute rest time between each sprint. The time was recorded using an iPhone 6 and MySprint app (Apple Inc., Cupertino, CA, USA). The reliability and validity of this method has been reported to be excellent [33]. The fastest time of the two sprints was used for analysis.

### V-cut test

In the change of direction test, players performed a 25-m sprint with four change directions of 45˚ 5 m each. The front foot was placed 0.5 m before the first timing gate. There were marks on the floor and cones, so subjects knew when to change of direction. For the trial to be valid, players had to pass the line, placed on the floor, with one foot completely at every turn. If the trial was considered a failed attempt, a new trial was allowed. The distance between each pair of cones was 0.7 m. Two trials were completed with a three-minute rest time between each trial. The fastest time was used for analysis. Time to completion was measured using a double beam photocell connected to a computer (Chronojump BoscoSystem, Barcelona, Spain) [34]. This test has previously demonstrated good reliability and validity [35].

### Analyses

Statistical analyses were performed using SPSS (Version 20 for Windows; SPSS Inc., Chicago, IL, USA). Descriptive statistics were derived (mean and standard deviation) for all the variables. Kolmogorov-Smirnov test was used to check the normality of the tested parameters. In addition, within-session reliability of test measures were analyzed using two way random intraclass correlation coefficient (ICC) with absolute agreement (95% confidence intervals) and coefficient of variation (CV). For interpretation, intraclass correlation coefficient (ICC) values were $> 0.9$ = excellent, 0.75–0.9 = good, 0.5–0.75 = moderate, and $< 0.5$ = poor [36] and CV values were considered acceptable if $< 10\%$ [37]. The number of subjects chosen was based on effect size 0.30 SD with an $\alpha$ level of 0.05 and power at 0.95 using G Power Software (University of Dusseldorf, Germany).

For the purpose of identifying inter-limb asymmetry between limbs, we also calculated the asymmetry index using the following formula [20,38,39] in the unilateral vertical and horizontal jumping and dynamic balance capacity: (Highest performing limb–Lowest performing

limb/Highest performing limb) ×100. The highest performing was defined as the side with the highest value of each task.

As the variables were normally distributed; student *t*-tests were performed to compare the inter-limb differences in the whole group and when differentiated by sex. In addition, independent-samples *t*-tests were used to determine possible differences in asymmetries (SEBT ANT, OLHT and SLCMJ) between males and females. Moreover, the magnitude of the difference in asymmetry between males and females were determined using Cohen's *d* effect sizes (ES) using the formula: $(\text{Mean}_{\text{male}} - \text{Mean}_{\text{female}})/\text{SD}_{\text{pooled}}$. Values were interpreted in line with suggestions from Hopkins et al. [40] where $< 0.20$ = trivial; 0.20–0.60 = small; 0.61–1.20 = moderate; 1.21–2.0 = large and $> 2.0$ = very large.

Pearsons correlations (*r*) was computed to compare asymmetry scores (SLCMJ, OLHT and SEBT ANT) and physical performance (30-m sprint performace, V-cut test, SLCMJ, OLHT and ANT SEBT). Statistical significance was established *a priori* at $p \leq 0.05$. Secondly, and differing by sex (i.e., exclusively males or females), the same correlation analysis was performed. The magnitude of Pearson correlation was interpreted in line with suggestions from Hopkins et al. (2009) where 0.0–0.1 = trivial; 0.1–0.3 = small; 0.3–0.5 = moderate; 0.5–0.7 = large; 0.7–0.9 = very large and 0.9–1 = almost perfect [41].

## Results

Mean and standard deviation for all the variables are shown in Table 2. Almost all the tests showed excellent within-session ICC values ($\geq 0.9$) and each test had acceptable consistency with all CV values $< 10\%$. Significant differences between the highest and lowest performing limbs were seen during the SLCMJ and OLHT ($p < 0.001$) for the total sample and when separated by sex. No significant differences between limbs were shown during the SEBT.

When comparing asymmetry values between sexes (t-test) there were no significant differences in vertical (p = 0.06) and horitzontal (p = 0.61) jumping tests. However, there were significant differences in asymmetry betweeen sexes in the ANT SEBT (*p* = 0.00), with greater

**Table 2. Mean test scores ± standard deviations and within-session reliability data for the star excursion balance test (SEBT), one legged hop test (OLHT) and single leg countermovement jump (SLCMJ), comparing the highest and lowest performing limbs during each test.**

| Test | Total (*n* = 81) | Males (*n* = 30) | Females (*n* = 51) | ICC (95% CI) | CV (%) |
|---|---|---|---|---|---|
| SEBT ANT-HPL (m) | 6.73 ± 10.9 | 68.2 ± 9.57 | 69.9 ± 6.1 | 0.86 (0.64–0.93) | 1.12 |
| SEBT ANT-LPL (m) | 6.87 ± 6.35 | 68.9 ± 6.91 | 69.3 ± 7.97 | 0.75 (0.65–0.81) | 1.53 |
| *p* | 0.37 | 0.51 | 0.74 | | |
| Asymmetry (%) | 13.6 ± 15.3 | 15.3 ± 18.8 | 11.3 ± 12.9 | | |
| OLHT-HPL (m) | 1.53 ± 0.15 | 1.68 ± 0.24 | 1.94 ± 0.15 | 0.89 (0.83–0.93) | 2.78 |
| OLHT-LPL (m) | 1.44 ± 0.18 | 1.58 ± 0.25 | 1.83 ± 0.17 | 0.94 (0.88–0.97) | 2.75 |
| *p* | < 0.001 | < 0.001 | < 0.001 | | |
| Asymmetry (%) | 5.99 ± 6.25 | 5.92 ± 4.61 | 6.03 ± 7.0 | | |
| SLCMJ-HPL (m) | 0.14 ± 0.02 | 0.16 ± 0.03 | 0.19 ± 0.03 | 0.96 (0.93–0.97) | 2.54 |
| SLCMJ-LPL (m) | 0.12 ± 0.02 | 0.14 ± 0.03 | 0.17 ± 0.03 | 0.95 (0.93–0.96) | 2.70 |
| *p* | < 0.001 | < 0.001 | < 0.001 | | |
| Asymmetry (%) | 11.3 ± 8.32 | 12.9 ± 10.4 | 10.8 ± 6.92 | | |
| 30 m (s) | 4.85 ± 0.29 | 4.67 ± 0.35 | 4.35 ± 0.20 | 0.94 (0.87–0.96) | 6.34 |
| V-cut (s) | 7.54 ± 0.31 | 7.18 ± 0.95 | 6.56 ± 1.30 | 0.96 (0.94–0.97) | 5.12 |

ICC = intraclass correlation coefficient; CI = confidence intervals; CV = coefficient of variation; ANT = anterior; HPL = highest performing limb; LPL = lowest performing limb; m = meter; s = second.

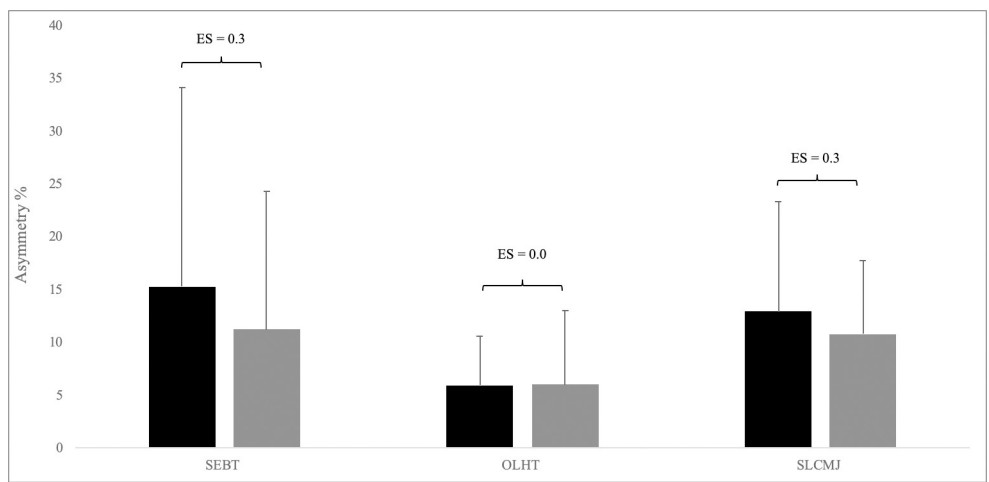

**Fig 1. Mean inter-limb asymmetry values ± standard deviations (error bars) for males (black bars) and females (grey bars) during the star excursion balance test (SEBT), one legged hop test (OLHT) and single leg countermovement jump (SLCMJ) tests.** ES = effect size.

asymetries in male population (Table 2). In addition, mean inter-limb asymmetry values for male and female athletes are shown in Fig 1. The SEBT showed between-limb differences of 15.3 ± 18.8% for males and 11.3 ± 12.9% for females, representing a small ES (0.25). The OLHT showed between-limb differences of 5.92 ± 4.61% for males and 6.03 ± 7.0% for females, representing a trivial ES (0.0). The SLCMJ showed between-limb differences of 12.9 ± 10.4% for males and 10.8 ± 6.92% for females, representing a small ES (0.3).

Pearson's correlations ($r$) between inter-limb asymmetry and test scores are shown in Table 3. Results showed significant ($p < 0.05$) but small ($r = 0.26$) relationships between SLCMJ asymmetries and 30 m sprint time for the total group. Significant negative correlations with small to moderate magnitude of correlation were also present between SLCMJ asymmetries and SLCMJ performance on the lowest performing limb for the total group ($p < 0.05$;

**Table 3. Pearson's correlations ($r$) between inter-limb asymmetry values and performance test scores for the total group and when separated by sex.**

| Asymmetry Index | 30 m | V-cut | OLHT (HPL) | OLHT (LPL) | SLCMJ (HPL) | SLCMJ (LPL) | SEBT (HPL) | SEBT (LPL) |
|---|---|---|---|---|---|---|---|---|
| *SEBT (%)* | | | | | | | | |
| Total ($n = 81$) | -0.08 | 0.01 | 0.16 | 0.20 | 0.03 | -0.01 | 0.09 | -0.10 |
| Male ($n = 30$) | -0.13 | 0.05 | 0.14 | 0.27 | -0.17 | -0.27 | 0.03 | 0.09 |
| Female ($n = 51$) | -0.02 | 0.05 | 0.23 | 0.24 | 0.13 | 0.11 | 0.08 | -0.14 |
| *OLHT (%)* | | | | | | | | |
| Total ($n = 81$) | -0.08 | 0.06 | -0.10 | -0.44† | -0.09 | -0.12 | -0.16 | -0.14 |
| Male ($n = 30$) | -0.13 | 0.15 | -0.44 | -0.56† | 0.12 | 0.09 | 0.30 | -0.31 |
| Female ($n = 51$) | -0.01 | 0.06 | 0.17 | -0.64† | 0.22 | 0.28 | -0.18 | -0.22 |
| *SLCMJ (%)* | | | | | | | | |
| Total ($n = 81$) | 0.26* | 0.01 | 0.16 | 0.12 | 0.11 | -0.26* | 0.06 | 0.18 |
| Male ($n = 30$) | 0.36 | 0.10 | 0.16 | 0.17 | 0.27 | -0.48† | 0.24 | 0.26 |
| Female ($n = 51$) | 0.14 | 0.10 | 0.08 | -0.02 | 0.06 | -0.30* | 0.09 | 0.15 |

† Significant at $p < 0.01$

* significant at $p < 0.05$

OLHT = one legged hop test; SLCMJ = single leg countermovement jump; SEBT = star excursion balance test; HPL = highest performing limb; LPL = lowest performing limb.

$r$ = -0.26), males ($p < 0.01$; $r$ = -0.48) and females ($p < 0.05$; $r$ = -0.30). Significant negative correlations with moderate and large margnitude were also present between OLHT asymmetries and OLHT performance on the lowest performing limb for the total group ($p < 0.01$; $r$ = -0.44), males ($p < 0.01$;$r$ = -0.56) and females ($p < 0.01$; $r$ = -0.64). In addition, no correlation were observed between SEBT ANT asymmetries and physical performance.

## Discussion

The aims of the present study were to examine the relationship between inter-limb asymmetries and physical performance in elite youth team-sports players, and to determine the difference in asymmetry between sexes in youth team sport athletes. Results showed that larger jump height asymmetries during the SLCMJ test were associated (small-to-moderate magnitude) with lower 30 m sprint performance when the entire sample were considered. The same asymmetry index was also associated reduced jump height on the lowest performing limb for the entire sample and when males and females were separated. Larger asymmetries during the OLHT were also associated with reduced distance during the OLHT, on the lowest performing limb for the whole sample and when separated by males and females.

One of the primary findings of this study was the significant relationship between jump height asymmetries from the SLCMJ and 30-m sprint time. However we have to be cautious with this finding because the magnitude of the correlation was small ($r$ = 0.26), these data indicate that larger jump height asymmetries were associated with slower sprint times for the whole group ($n$ = 81) (Table 3). This is in accordance with recent literature on this topic. For example, larger jump height asymmetries during the SLCMJ have been associated with slower acceleration performance ($r$ = 0.49 to 0.59; $p < 0.05$) in youth female soccer players (7). Furthermore, jump height asymmetries (again from the SLCMJ), have also been correlated with reduced acceleration ($r$ = 0.54 to 0.87; $p < 0.05$) performance in academy male soccer players (35). Thus, the findings from the present study are in agreement with recent literature pertaining to asymmetry and linear speed. In contrast, in the present study, no significant associations were evident between jump height asymmetry and total time during the V-cut test. This is also in agreement with previous research. Lockie et al. (16) and Dos'Santos et al. (17) both showed no association between jump height and distance asymmetry from the SLCMJ and single and triple hop tests and COD (505) performance in male collegiate athletes. However, this is in contrast to previous research wich has shown realtionship between major inter-limb asymmetries and COD speed decrements [3,42]. Thus, the lack of consensus regarding the impact of inter-limb asymmetries on COD performance could be explained because of differences in the assessement methods, asymmetry calculations and athlete population.

Another main finding of this study is that larger jump height asymmetries (vertical and horizontal directions) are associated with diminished jump performance in the weaker leg and appear to be direction-specific. However, these results do have some limitations that practitioners should be aware of. The significant correlation between asymmetry and the weaker leg is to be expected, given that asymmetry is ratio of both the stronger and weaker limbs. Therefore, it is suggested that correlations with ratios should be made with independent variables (e.g., jump height asymmetry vs. speed or CODS). Despite this, these results are in accordance with Bishop et al. [7] who also showed that larger jump height and distance asymmetries (from the SLCMJ and triple hop tests) were also associated with reduced jump performance during the SLCMJ and horizontal hop tests respectively. In this case, they found decrements in performance in both limbs, contrary to our study, where significant correlations were noted only in the weaker limb. The differences in results between studies could be explained because Bishop et al. (7) presented the data as left vs. right (not stronger vs. weaker). Given these findings, it

seems logical to consider the weaker limb within the context of a "window of opportunity" for improving physical capacity in the desired task (9). It has been suggested that existing asymmetries can be reduced if both bilateral and unilateral training methods are programmed concurrently [43]. However, recent research has shown that double the volume on the weaker limb may be effective at reducing between-limb differences during jump tasks [44].

The relationship between larger jumping asymmetries and impaired performance in our study is supported by the significant differences between lower-limbs founded in vertical and horizontal tests, but not in the balance test. Despite both SLCMJ and OLHT were sensitive to detect significant inter-limb differences, vertical jumping produced considerably greater asymmetries (11.33 ± 8.32) than horizontal jumping (5.99 ± 6.25). The largest inter-limb asymmetry detected from the SLCMJ in this study is in agreement with the 10–15% threshhold of potential risk of injury described by previous literature [2,5,6,10] and is likely the most sensitive task for detecting between-limb asymmetries from the test battery used in the present study. Strength and conditioning coaches can use this information to guide neuromuscular performance programs in youth team-sports players. However, it is difficult to establish a true threshold for asymmetry given the task-specific nature of this concept (2,3,7,10,16,17). The OLHT asymmetry values of 5.99% can be considered small (38) and similar to those reported by Hewitt et al. (2) (4.6%) and Bishop et al. (7) (~6%). In addition, asymmetry values shown on the SEBT ANT (13.59%) were notably larger than the ~5% value reported by Overmoyer et al. (37). The fact that in our study the ASI values in the SEBT ANT were higher than those shown in the literature may be due to the sample used. While Overmoyer et al. (37) used healthy, recreationally active young adults, in our case we used young elite sport team athletes where the ankle sprain is usually one of the most common injuries [45]. One of the long-term consequences of ankle injuries is the limitation of ankle dorsal flexion [46,47], which is related to worse SEBT ANT values in the anterior direction [25].

In the present study we also evaluate for the presence of inter-limb asymmetries in elite youth team sports players in the whole group and when differentiating by sex. Only unilateral vertical and horizontal jumping asymmetries were significant in the whole group and when differentiating by sex. This is in accordance with several studies that studied jump performance asymmetries in youth athletes [7,11,48]. Conversely, there were no significant differences between limbs in the dynamic balance test. This is in accordance with the study of Overmoyer et al. (37), where no significant between-limb differences were found for the SEBT when comparing between the kicking and non-kicking leg in active young athletes. When our findings are considered within this context, it may suggest that the assessment of asymmetries should include more functional actions characterized by the sport physical demands (e.g., jumping).

A secondary objective of the present study was to evaluate for the presence of difference in asymmetry between sexes in youth team sport athletes. Some authors suggested that the inter-limb asymmetry is frequently greater in female compared to male athletes in relation to strength, coordination and postural control [49]. There are many studies reporting a higher ankle and knee joint injury incidence in women [50,51]. With respect to the latter, women show a greater number of specific injuries such as anterior knee pain [52], the ACL rupture [5] and ankle sprains [53]. This greater number of injuries in female athletes has been related to neuromuscular factors, including the physical capacity predominance of one leg over the other [22,54]. Despite this, our hypotheses was that the level of asymmetry would be more pronounced in females than that in males. However, our hypothesis has been disproven with no significant diffferences between groups in unilateral jumping tasks. Our results are also in disagreement with Bailey et al. (39) who showed that female athletes depicted greater between-limb differences in bilateral jumping compared with male counterparts. Whilst the difference

in results could be explained by the chosen test methods (i.e., unilateral in the present study), the variability of asymmetry must also be considered within the context of these findings. Fig 1 shows the very large SD associated with inter-limb asymmetry relative to the group mean value; thus, detecting significant between-group differences in asymmetry is challenging. However, this is complimented by the ES statistic showing trivial to small differences were present between sexes. In addition, and also contrary to our hypothesis, we found that males had significantly greater inter-limb asymmetries than females during the dynamic balance test. This is again in contrast to Holden et al. (40), who showed no differences in dynamic postural stability performance (composite SEBT) between young adolescent male and female secondary school athletes. Although challenging to fully explain, these differences could be explained by the wide variety of sporting backgrounds and abilities used in Holden et al's study (40), noting that the sample was made up of secondary school children. The present study used competitive team sport athletes and although jumping was superior at detecting larger magnitudes of asymmetry, the SEBT was able to differentiate between sexes. Despite this, our results should be considered with caution due to the small sample of males (n = 30) compared to females (n = 51).

Despite the usefulness of these findings, the present study has some limitations which must be acknowledged. Firstly, the effect of asymmetries on physical performance should be studied according to the different maturational stages and sex and there is a paucity of research investigating this conclusively. Read et al. [20] observed that landing force asymmetries during the SLCMJ increased with maturation. In addition, inter-limb asymmetries have been shown to be highly task-specific [11,16] and, in this study, we only studied jumping height (with a contact mat) and dynamic balance asymmetries. So, it would have been interesting to include the analysis of asymmetries during change of direction tasks, a key factor in team sports performance. In addition, future research should look to analyze asymmetry from a longitudinal perspective.

## Conclusions

Given the associations between asymmetries and reduced sprint speed and jumping performance, it is suggested that strength and conditioning training interventions should consider the reduction of inter-limb asymmetry in youth team-sport athletes. Moreover, higher asymmetries were associated with diminished jump performance in the weaker leg and appear to be direction-specific. Thus, the fact of having inter-limb asymmetries gives us an opportunity to train and improve from the weaker leg. In addition, practitioners should also consider strengthening the weaker limb during sport-specific skills as well.

For example, in a basketball game, performing a "lay-up" on the weaker side (weaker leg) successfully can benefit the performance of the player. Thus, the concept of a window of opportunity for the weaker limb should also be considered within the context of sport skills too.

## Supporting information

**S1 File. STROBE checklist of the study.**
(DOCX)

**S2 File. Clinical studies checklist.**
(DOCX)

## Acknowledgments

We are grateful to all the study subjects for their participation

## Author Contributions

**Conceptualization:** Azahara Fort-Vanmeerhaeghe, Chris Bishop, Bernat Buscà, Joan Aguilera-Castells, Oliver Gonzalo-Skok.

**Data curation:** Azahara Fort-Vanmeerhaeghe, Jordi Vicens-Bordas.

**Formal analysis:** Azahara Fort-Vanmeerhaeghe, Chris Bishop.

**Investigation:** Azahara Fort-Vanmeerhaeghe.

**Methodology:** Azahara Fort-Vanmeerhaeghe, Bernat Buscà, Oliver Gonzalo-Skok.

**Project administration:** Azahara Fort-Vanmeerhaeghe.

**Resources:** Azahara Fort-Vanmeerhaeghe.

**Software:** Azahara Fort-Vanmeerhaeghe.

**Supervision:** Azahara Fort-Vanmeerhaeghe, Chris Bishop.

**Validation:** Azahara Fort-Vanmeerhaeghe.

**Visualization:** Azahara Fort-Vanmeerhaeghe.

**Writing – original draft:** Azahara Fort-Vanmeerhaeghe, Oliver Gonzalo-Skok.

**Writing – review & editing:** Azahara Fort-Vanmeerhaeghe, Chris Bishop, Joan Aguilera-Castells, Jordi Vicens-Bordas, Oliver Gonzalo-Skok.

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
