## [Decision Letter · Decision Letter 0]

28 Aug 2019

PONE-D-19-17934

Neuromuscular asymmetries are associated with decrements in physical performance in youth elite team sports athletes

PLOS ONE

Dear Mrs Fort,

Thank you for submitting your manuscript to PLOS ONE. After careful consideration, we feel that it has merit but does not fully meet PLOS ONE’s publication criteria as it currently stands. Therefore, we invite you to submit a revised version of the manuscript that addresses the points raised during the review process.

We would appreciate receiving your revised manuscript by Oct 12 2019 11:59PM. To enhance the reproducibility of your results, we recommend that if applicable you deposit your laboratory protocols in protocols.io, where a protocol can be assigned its own identifier (DOI) such that it can be cited independently in the future. For instructions see: http://journals.plos.org/plosone/s/submission-guidelines#loc-laboratory-protocols

We look forward to receiving your revised manuscript.

Kind regards,

Andreas Mierau, PhD

Academic Editor

PLOS ONE

Journal Requirements:

2. Could you please state in your methods section on which dates the various tests were performed?

Reviewers' comments:

Reviewer's Responses to Questions

**Comments to the Author**

1. Is the manuscript technically sound, and do the data support the conclusions?

Reviewer #1: Partly

Reviewer #2: Yes

2. Has the statistical analysis been performed appropriately and rigorously? 

Reviewer #1: Yes

Reviewer #2: Yes

3. Have the authors made all data underlying the findings in their manuscript fully available?

Reviewer #1: Yes

Reviewer #2: Yes

4. Is the manuscript presented in an intelligible fashion and written in standard English?

Reviewer #1: Yes

Reviewer #2: Yes

5. Review Comments to the Author

Reviewer #1: General comments

This is a cross sectional study aimed at examining the associations between neuromuscular asymmetries and physical performance in youth male and female athletes. While the topic sounds interesting, and it is I think, the practical applications do not fit well with the main outcome of the study. The discussion needs deep revision as it looks quite superficial. Also, there are several spelling mistakes that require the attention of authors. Additional comments and suggestions are listed below.

Abstract

Line 24-26 In the introduction you mentioned that the second purpose of this study was to examine the between-sexes differences. This has to be outlined here. Also the respective outcomes need to be displayed in the results.

No age specific correlation?

29 V-cut: Either big or small “V”.

33-34: I would suggest to specify the magnitude of the correlation across the abstract

39-41 It would be better to stick to the exact measures of physical fitness undertaken in this study. These are sprint speed and jumping. Physical fitness is too vague and concluding that jumping asymmetries were correlated with decrements in physical fitness from a general perspective is not supported by the findings making the statement inaccurate.

Introduction

The introduction is comprehensively written. The rational of the study is clearly emphasized.

53 something wrong here in terms of writing

54-57 To be able to easier grasp the meaning of this sentence, I suggest to split it into two.

107-109 The second objective is unclear to me. I would mainly focus on the between-sexes differences given that looking for neuromuscular asymmetries seems not to be an objective per se. The existence of neuromuscular asymmetries has been well evidenced.

110-112 please add the corresponding literature based on which you built the hypothesis.

Methods

117 In a group of youth instead

The term elite needs to be further clarified.

119 change of direction speed instead and throughout the manuscript

124 Please clarify how maturity level was estimated?

127 Please refer to my comment above. What elite refers to?

130 8-10 sessions per week with 90-120 min per session instead

150-151 "Testing protocols were performed in two separate days with 24 to 48 h in-between. In the first day ***** were conducted while in the second day **** were carried out" INSTEAD

188 Do you mean "were"? and even with "were" the meaning still unclear. Please, clarify.

207 Trial?

210 Single or double beam system?

More details on how reliability was assessed are needed in the methods section. For instance, the number of days that separated the test and retest sessions etc

214 Analyses instead

215 were instead

220 coefficient of variation instead

222 How can this be poor an acceptable at the same time? It does not make sense.

the cut off value for good absolute reliability is usually <5%.

230 Effect size tell us about the magnitude of the difference (if any). However, to get to know whether a difference exists or not you may need to calculate p value.

235 what was the scale used to judge the magnitude of the correlation coefficient values?

238-239 Unclear, are you talking about correlation analysis here. Please, clarify.

239-243 Ok now I see. this has to be moved above.

Results

249 Please add the corresponding effect size value consistently.

250 No significant differences between the dominant and non-dominant limb were shown instead

Table 2 Something wrong here. I don't expect an anterior reach distance in the SEBT of 68 m. Surely, this is expressed in cm.

Would be better to round the values to one single decimal after the dot. Please, do this consistently throughout the manuscript.

Bottom table 2: meter instead, second instead

259-262 I would rather report on the significance (or not) of the difference and then the respective effect size.

265-268 you may want to simply note for example "figure one near here" and associate the full title to the respective figure below.

271 How about the magnitude of the correlation: small, large, very large? Please specify in a consistent manner throughout the manuscript.

272-275 I think this a kind of expected outcome as the asymmetry level is dependent on the weaker side performance. However, the small magnitude of the association surprises me.

275-278 Again this is quite expected.

Discussion

285 and to determine instead

294-297 reads awkward. Please, revise.

303 “no significant findings” association instead

306 could you specify the exact CoD test?

307 Could you speculate on the reasons underlying the significant association between jump asymmetries and sprint speed? Also could you argue about the absence of correlation with CoD test?

315-317 Why is that? could you develop further on this?

317-319 Unclear / incomplete sentence

333-335 I'm not sure about the need of such statement having in mind the small sample size included in this study. I don't think that from a sample of 30 male subjects we can have robust normative values. Idem for females.

338- 340 I think that the essence of a discussion is not to simply state that the current findings are larger or smaller compared to previous results but rather to explain the potential reasons behind. In terms of this particular statement, this may have something to do with the testing population, testing protocols themselves, training expertise etc. Please, try to expand more your statements and discuss about the potential reasons for discrepancies/agreement.

362-365 This may have something to do with the small sample size of males engaged in the comparison. Other than this, why authors expected to have larger asymmetries with females than males? any physiological explanations for this?

367-375 Authors explanations are somehow superficial and do not go deep into the point. The discussion needs to benefit from a substantial revision to make it more informative about the current findings in lights of the previous outcomes. More importantly, readers would expect deeper explanations (speculations) related to the different controversies with the previous literature.

Conclusions

The concept of window of opportunity was previously questioned and there are a lot of concerns around it. This is basically because all measures of physical fitness are trainable throughout childhood and there is no restriction to a specific period. In the context of this study, it may be better to simply recommend to further develop neuromuscular performance of the weaker limb to mitigate the difference with the stronger side and accordingly the asymmetry level.

Reviewer #2: Abstract

- Please add some background information about the topic.

- Why did you not mention sex differences?

Introduction

- l.53: "with" or "to", but not both ...

- ll.62/108: What are neuromuscular asymmetries? Are you really measuring neuromuscular parameters?

- l.79: COD is clear, but what is "CODS"?

- ll.81/82: "505 on both limbs"?

- ll.94/95: At this stage it is surprising that you introduce sex differences because neither title nor abstract deal with this issue.

Methods

- l.117: "group youth elite team-sports players"? please rewrite

- l.125: Please specify the distribution of team sports within sexes.

- Table 1: Please add information about how you assessed "years post-PHV".

- l.157: What does the SEBT ANT measures? Which abilities and muscular capacities are needed to perform well?

- l.176: What did you do if partcipants bend their knees during landing? This might have had an impact on flight time and thus on jumping height.

- l.180: Why did you execute the SLCMJ only with the preferred leg?

- l.186: I do not understand why the OLHT should estimate "dynamic stability". Please expand.

Results

- Please consider to indicate just one decimal place for all values presented except for sprint times.

- l.254: typos

- Table 3: Please do not use ** for significance at p<0.01.

Discussion

- ll.286/287: Be more reserved as those associations are just small (R²=~7%).

- ll.289: What is the practical importance of the 19% relationship between OLHT asymmetries and OLHT performance?

- l.293: A primary result which comprises a 7% relation between two parameters is poor ...

- ll. 297-301: Link theses findings to your results!

- ll.328-329: Why? What is the reason/mechanism for this phenomenon?

- ll.380ff: Please add that you used a contact mat to determine jumping height. Why did you not use an isokinetic dynamometer to assess interlimb asymmetries?

- l.382: How do you measure task specifity?

Conclusions

- Please rewrite the whole paragraph. Concentrate on your findings and major implications.

---

## [Author Response · Author response to Decision Letter 0]

12 Oct 2019

Response to Reviewers

Reviewer #1: General comments

- This is a cross sectional study aimed at examining the associations between neuromuscular asymmetries and physical performance in youth male and female athletes. While the topic sounds interesting, and it is I think, the practical applications do not fit well with the main outcome of the study. The discussion needs deep revision as it looks quite superficial. Also, there are several spelling mistakes that require the attention of authors. Additional comments and suggestions are listed below.

Abstract

- Line 24-26 In the introduction you mentioned that the second purpose of this study was to examine the between-sexes differences. This has to be outlined here. Also the respective outcomes need to be displayed in the results.

Added in the abstract: 

“A secondary objective was to evaluate the presence of between-sexes differences in inter-limb neuromuscular asymmetries¬¬ in elite youth team sports players.” 

“In addition, when comparing asymmetry values between sexes there were no significant differences in vertical (p = 0.06) and horizontal (p = 0.61) jumping tests. However, there were significant differences in asymmetry between sexes in the ANT SEBT (p = 0.04)”

- No age specific correlation? 

Although this wasn’t one of the objectives of our study it would be interesting to perform this analysis in future studies with a bigger range of age. In our case, when we compared groups u-14/16 with group u16-18 we found no significant differences.

- 29 V-cut: Either big or small “V”.

Changed to “V-cut test”

- 33-34: I would suggest to specify the magnitude of the correlation across the abstract

The qualitative magnitude (large, small…) was added across the text (See Methods and results). In the abstract we only included the numeric magnitude.

- 39-41 It would be better to stick to the exact measures of physical fitness undertaken in this study. These are sprint speed and jumping. Physical fitness is too vague and concluding that jumping asymmetries were correlated with decrements in physical fitness from a general perspective is not supported by the findings making the statement inaccurate.

Changed: “In conclusion, the current study indicated that jumping asymmetries were associated with decrements in sprint speed and jumping performance.”

Introduction

- The introduction is comprehensively written. The rational of the study is clearly emphasized.

- 53 something wrong here in terms of writing

“Between-limb differences in power and strength have been purported as important risk factors for sport injuries 1,2 and, in some instances, have been associated with to decrements in sporting performance 3,4.”

- 54-57 To be able to easier grasp the meaning of this sentence, I suggest to split it into two.

Changed. In any given task, the reduced physical capacity of the weaker limb to both produce and absorb force is likely to increase the risk of injury. This is because it is likely to exceed its “tolerance capacity” sooner than the stronger limb when repeated high intensity actions are considered [5,8].

- 107-109 The second objective is unclear to me. I would mainly focus on the between-sexes differences given that looking for neuromuscular asymmetries seems not to be an objective per se. The existence of neuromuscular asymmetries has been well evidenced.

We agree. We clarified: “A secondary objective was to evaluate the presence of between-sexes differences in inter-limb neuromuscular asymmetries¬¬ in elite youth team sports players.”

110-112 please add the corresponding literature based on which you built the hypothesis.

Added (19-21).

Methods

117 In a group of youth instead

Changed.

The term elite needs to be further clarified.

This group is recognized as elite by the Spanish Sports Council. 

We added in the participants section: “All the participants train and study in the same high performance sports center, in Esplugues de Llobregat (Joaquim Blume Residence).”

119 change of direction speed instead and throughout the manuscript

Changed.

124 Please clarify how maturity level was estimated?

Added. “Biological maturation was calculated in a noninvasive manner using a regression equation comprising measures of age, body mass, standing height, and sitting height 5.”

127 Please refer to my comment above. What elite refers to?

Clarified below. 

130 8-10 sessions per week with 90-120 min per session instead

Changed.

150-151 "Testing protocols were performed in two separate days with 24 to 48 h in-between. In the first day ***** were conducted while in the second day **** were carried out" INSTEAD

Changed. “Testing was performed in two separate days with 24-48 hours in-between. In the first day were conducted SEBT, SLCMJ and OLHT while in the second day 30-m sprint and V-cut test.”

207 Trial?

Changed.

210 Single or double beam system?

Added. “The fastest time was used for analysis. Time to completion was measured using a double beam photocell connected to a computer”.

More details on how reliability was assessed are needed in the methods section. For instance, the number of days that separated the test and retest sessions etc

We added more details on reliability of the study. 

“One week before data collection, all participants were familiarized with performance tests procedures enabling them to practice each test between 2-5 times.”

“Testing was performed in two separate days with 24-48 hours in-between. In the first day were conducted SEBT, SLCMJ and OLHT while in the second day 30-m sprint and V-cut test.”

There is not a retest session. “In addition, within-session reliability of test measures were analyzed using two way random intraclass correlation coefficient (ICC) with absolute agreement (95% confidence intervals) and coefficient of variation (CV).”

214 Analyses instead

Changed.

215 were instead

Changed.

220 coefficient of variation instead

Added.

222 How can this be poor an acceptable at the same time? It does not make sense.

the cut off value for good absolute reliability is usually <5%.

Sorry for the mistake. Changed. “For interpretation, intraclass correlation coefficient (ICC) values were > 0.9 = excellent, 0.75-0.9 = good, 0.5-0.75 = moderate, and < 0.5 = poor and acceptable 6 and CV values were considered acceptable if < 10% [35] “ We added Cormack et al. reference [35].

230 Effect size tell us about the magnitude of the difference (if any). However, to get to know whether a difference exists or not you may need to calculate p value.

Yes, totally agree. Our text wasn’t clear. We reorder the statistical section to clarify the analysis. “In addition, independent-samples t-tests were used to determine possible differences in asymmetries (SEBT ANT, OLHT and SLCMJ) between males and females. Moreover, differences in asymmetry between males and females were determined using Cohen’s d effect sizes (ES) using the formula: (Meanmale – Meanfemale)/SDpooled. Values were interpreted in line with suggestions from Hopkins et al. [38] where < 0.20 = trivial; 0.20-0.60 = small; 0.61-1.20 = moderate; 1.21-2.0 = large and > 2.0 = very large.”

235 what was the scale used to judge the magnitude of the correlation coefficient values?

Added. “The magnitude of Pearson correlation was interpreted in line with suggestions from Hopkins et al. (2009) where 0.0-0.1 = trivial; 0.1-0.3 = small; 0.3-0.5 = moderate; 0.5-0.7 = large; 0.7-0.9 = very large and 0.9-1 = almost perfect [40].”

238-239 Unclear, are you talking about correlation analysis here. Please, clarify.

Clarified. Secondly, and differing by sex (i.e., exclusively males or females), the same correlation analysis was performed.

239-243 Ok now I see. this has to be moved above.

Moved.

Results

249 Please add the corresponding effect size value consistently.

We have only calculated Cohen’s d effect sizes (ES) to assess interlimb differences between sexes, although if the reviewer considers it necessary we can add it.

250 No significant differences between the dominant and non-dominant limb were shown instead

Modified. “No significant differences between limbs were shown during the SEBT”

Table 2 Something wrong here. I don't expect an anterior reach distance in the SEBT of 68 m. Surely, this is expressed in cm.

Sorry for the mistake. Changed to meters.

Would be better to round the values to one single decimal after the dot. Please, do this consistently throughout the manuscript.

Changed to one single decimal throughout the manuscript.

Bottom table 2: meter instead, second instead

Modified.

259-262 I would rather report on the significance (or not) of the difference and then the respective effect size.

Thanks for the suggestion. We reorder the text reporting the significance first and the effect size later. “When comparing asymmetry values between sexes (t-test) there were no significant differences in vertical (p = 0.1) and horitzontal (p = 0.6) jumping tests. However, there were significant differences in asymmetry betweeen sexes in the ANT SEBT (p = 0.0), with greater asymetries in male population (Table 2). In addition, mean inter-limb asymmetry values for male and female athletes are shown in Fig 1. The SEBT showed between-limb differences of 15.3 ± 18.8% for males and 11.3 ± 12.9% for females, representing a small ES (0.25). The OLHT showed between-limb differences of 6.9 ± 4.6% for males and 6.0 ± 7.0% for females, representing a trivial ES (0.0). The SLCMJ showed between-limb differences of 12.9 ± 10.4% for males and 10.8 ± 6.9% for females, representing a small ES (0.3).”

265-268 you may want to simply note for example "figure one near here" and associate the full title to the respective figure below.

Added.

271 How about the magnitude of the correlation: small, large, very large? Please specify in a consistent manner throughout the manuscript.

We rewrote the results and also added it in the discussion section. “Results showed significant (p ˂ 0.05) but small (r = 0.3) relationships between SLCMJ asymmetries and 30 m sprint time for the total group. Significant negative correlations were present between SLCMJ asymmetries and SLCMJ performance on the lowest performing limb for the total group (r = -0.3; p < 0.05), males (r = -0.5; p < 0.01) and females (r = -0.3; p < 0.05). However, the magnitudes of the correlations between SLCMJ asymmetries and SLCMJ performance on the lowest performing limb were small to moderate for the total group (r = -0.3), females (r = -0.3) and males (r = -0.5). Significant negative correlations with moderate and large magnitude were also present between OLHT asymmetries and OLHT performance on the lowest performing limb for the total group (p < 0.01; r = -0.5), males (p < 0.01; r = -0.6) and females (p < 0.01; r = -0.6).”

272-275 I think this a kind of expected outcome as the asymmetry level is dependent on the weaker side performance. However, the small magnitude of the association surprises me.

We are presenting those results from the statistical analysis.

275-278 Again this is quite expected.

Thank you for the feedback and because the reviewer did not request any amendments, we have not made any here.

Discussion

285 and to determine instead

Added. 

294-297 reads awkward. Please, revise.

 “Despite we have to be cautious with this finding because the magnitude of the correlation was small (r = 0.3), these data indicate that larger jump height asymmetries were associated with slower sprint times for the whole group (n = 83) and is in accordance with recent literature on this topic.”

303 “no significant findings” association instead

Added. 

306 could you specify the exact CoD test?

Specified “(505)”.

307 Could you speculate on the reasons underlying the significant association between jump asymmetries and sprint speed? Also could you argue about the absence of correlation with CoD 

That because it is associative analysis, it is impossible to come up with justified reasons in line with good science practice; thus, any speculation that isn’t entirely obvious doesn’t really add to the manuscript; rather, it makes things more unclear.

315-317 Why is that? could you develop further on this? 317-319 Unclear / incomplete sentence

We clarified. “Given these findings, it seems logical to consider the weaker limb within the context of a “window of opportunity” for improving physical capacity in the desired task.”

333-335 I'm not sure about the need of such statement having in mind the small sample size included in this study. I don't think that from a sample of 30 male subjects we can have robust normative values. Idem for females.

Changed. “Strength and conditioning coaches can use this information as normative data to guide neuromuscular performance programs in youth team-sports players.”

338- 340 I think that the essence of a discussion is not to simply state that the current findings are larger or smaller compared to previous results but rather to explain the potential reasons behind. In terms of this particular statement, this may have something to do with the testing population, testing protocols themselves, training expertise etc. Please, try to expand more your statements and discuss about the potential reasons for discrepancies/agreement.

We added. “The fact that in our study the ASI values in the SEBT ANT were higher than those shown in the literature may be due to the sample used. While Overmoyer et al. (37) used healthy, recreationally active young adults, in our case we used young elite sport team athletes where the ankle sprain is usually one of the most common injuries [43]. One of the long-term consequences of ankle injuries is the limitation of ankle dorsal flexion [44,45], which is related to worse SEBT ANT values in the anterior direction [23].”

362-365 This may have something to do with the small sample size of males engaged in the comparison. 

Agree. Added in the discussion. “Despite this, our results should be taken into account with caution due to the small sample of males (n�30) compared to females (n�51).”

Other than this, why authors expected to have larger asymmetries with females than males? any physiological explanations for this?

Added to the discussion. “Some authors suggested that the inter-limb asymmetry is frequently greater in female compared to male athletes in relation to strength, coordination and postural control [47]. There are many studies reporting a higher ankle and knee joint injury incidence in women [48,49]. With respect to the latter, women show a greater number of specific injuries such as anterior knee pain [50], the ACL rupture [5] and ankle sprains [51]. This greater number of injuries in female athletes has been related to neuromuscular factors, including the physical capacity predominance of one leg over the other [20,52].”

367-375 Authors explanations are somehow superficial and do not go deep into the point. The discussion needs to benefit from a substantial revision to make it more informative about the current findings in lights of the previous outcomes. More importantly, readers would expect deeper explanations (speculations) related to the different controversies with the previous literature.

Modified all the paragraph.

Conclusions

The concept of window of opportunity was previously questioned and there are a lot of concerns around it. This is basically because all measures of physical fitness are trainable throughout childhood and there is no restriction to a specific period. In the context of this study, it may be better to simply recommend to further develop neuromuscular performance of the weaker limb to mitigate the difference with the stronger side and accordingly the asymmetry level.

This concept was used in the last review of asymmetries and performance (Sean Maloney, 2018, JSCR). However, we clarified this concept in the conclusion section. “Given the associations between asymmetries and reduced sprint speed and jumping performance, it is suggested that strength and conditioning training interventions should consider the reduction of inter-limb asymmetry in youth team-sport athletes. Moreover, higher asymmetries were associated with diminished jump performance in the weaker leg and appear to be direction-specific. Thus, the fact of having inter-limb asymmetries gives us an opportunity to train and improve from the weaker leg. In addition, practitioners should also consider strengthening the weaker limb during sport-specific skills as well. For example, in the present study basketball athletes are frequently required to perform “lay-ups” which may occur on the weaker limb. Thus, the concept of a window of opportunity for the weaker limb should also be considered within the context of sport skills too.”

Reviewer #2: Abstract

- Please add some background information about the topic.

Added: Actually, there is scarce literature looking for the relationship between inter-limb neuromuscular asymmetries and performance in youth elite team sports.

- Why did you not mention sex differences?

Added in the abstract: 

“A secondary objective was to evaluate the presence of between-sexes differences in inter-limb neuromuscular asymmetries¬¬ in elite youth team sports players.” 

“In addition, when comparing asymmetry values between sexes there were no significant differences in vertical (p = 0.1) and horizontal (p = 0.6) jumping tests. However, there were significant differences in asymmetry between sexes in the ANT SEBT (p = 0.04).”

Introduction

- l.53: "with" or "to", but not both ...

Thanks, changed.

“Between-limb differences in power and strength have been purported as important risk factors for sport injuries 1,2 and, in some instances, have been associated with to decrements in sporting performance 3,4.”

- ll.62/108: What are neuromuscular asymmetries? Are you really measuring neuromuscular parameters?

We agree that as we measure indirectly neuromuscular characteristics, however we follow nomenclature specified in our previous publications.

Madruga-Parera, M., Romero-Rodríguez, D., Bishop, C., Beltran-Valls, M. R., Latinjak, A. T., Beato, M., & Fort-Vanmeerhaeghe, A. Effects of maturation on lower limb neuromuscular asymmetries in elite youth tennis players. Sports, 2019, 7(5), 106. https://doi.org/10.3390/sports7050106

Fort-Vanmeerhaeghe, A Gual G, , Romero-Rodriguez D, Unnithan Viswanath. Lower Limb Neuromuscular Asymmetry in Volleyball and Basketball Players. Journal of Human Kinetics 2016; 50(1):135-143.

Fort-Vanmeerhaeghe A, Montalvo A, Sitjà-Rabert M, Kiever A, Myer G. Neuromuscular asymmetries in the lower limbs of elite female youth basketball players and the application of the skillful limb model of comparison. Physical Therapy in Sport 2015; XXX: 1-7. 

- l.79: COD is clear, but what is "CODS"?

Change to “COD”.

- ll.81/82: "505 on both limbs"?

Deleted. “Specifically, jump height asymmetry was correlated with 30m (r = 0.58; p < 0.05) and 505 on both limbs (r = 0.52-0.66; p < 0.05), and reactive strength index asymmetry was correlated with 10m (r = 0.52; p < 0.05) and 505 on both limbs (r = 0.54-0.55; p < 0.05).”

- ll.94/95: At this stage it is surprising that you introduce sex differences because neither title nor abstract deal with this issue.

Introduced in the abstract as mentioned below.

Methods

- l.117: "group youth elite team-sports players"? please rewrite

Rewrited.

“in a group of youth elite team-sports players”

- l.125: Please specify the distribution of team sports within sexes.

Specified.” Eighty-one competitive team-sports players, specifically basketball (22 females), volleyball (14 females and 15 males) and handball (15 females and 15 males), volunteered to participate in this study with athletes tested in the pre-season period.”

- Table 1: Please add information about how you assessed "years post-PHV".

Added. 

Added. “Biological maturation was calculated in a noninvasive manner using a regression equation comprising measures of age, body mass, standing height, and sitting height 5.”

- l.157: What does the SEBT ANT measures? Which abilities and muscular capacities are needed to perform well?

Added. “The SEBT ANT measurement was used to asses dynamic balance [23], which is defined as the capacity to maintain center of mass over a fixed base of support under a challenge; specifically, motion of other limbs and body segments, or unanticipated disturbance to supporting surface [24]. ”

- l.176: What did you do if participants bend their knees during landing? This might have had an impact on flight time and thus on jumping height.

Any participant bends their knees. Agree with that comment.

- l.180: Why did you execute the SLCMJ only with the preferred leg?

We used the protocol used in our previous studies.

- l.186: I do not understand why the OLHT should estimate "dynamic stability". Please expand.

Dynamic stability is defined as the ability to maintain equilibrium during dynamic actions (the body is under some kind of displacement), bringing the requirement of balance to the kind of joint stability involved in sport-specific skills. Here some references (Noyes et al, 1996; Myer et al, 2011; Munro et al., 2012). We can add to the text if the reviewer considers it necessary.

Results

- Please consider to indicate just one decimal place for all values presented except for sprint times.

Changed to one decimal through all the text, as the other reviewer suggested.

- l.254: typos

Modified.

- Table 3: Please do not use ** for significance at p<0.01.

Changed to †.

Discussion

- ll.286/287: Be more reserved as those associations are just small (R²=~7%).

Agree. We changed.

“Results showed a tendency that larger asymmetries during the SLCMJ test were associated with slower 30 m sprint times when the entire sample were considered and reduced jump height on the lowest performing limb for the whole sample, males and females.”

- ll.289: What is the practical importance of the relationship between OLHT asymmetries and OLHT performance?

Explained and discussed below. 

- l.293: A primary result which comprises a 7% relation between two parameters is poor ...

Clarified. “Despite we have to be cautious with this finding because the magnitude of the correlation was small (r = 0.3), these data indicate that larger jump height asymmetries were associated with slower sprint times for the whole group (n = 83) and is in accordance with recent literature on this topic.”

- ll. 297-301: Link theses findings to your results!

Table added to the text. (Table 3). 

- ll.328-329: Why? What is the reason/mechanism for this phenomenon?

We do not know the reason, this is an associative study, but it coincides with previous studies described in the discussion.

- ll.380ff: Please add that you used a contact mat to determine jumping height. 

Added. In addition, neuromuscular asymmetries have been shown to be highly task-specific [11,16] and, in this study, we only studied jumping height (with a contact mat) and dynamic balance asymmetries. 

Why did you not use an isokinetic dynamometer to assess interlimb asymmetries?

It’s simply, for economic resources.

- l.382: How do you measure task specifity?

In this study and most publications, asymmetry is measured in tests that are close to fundamental motor skills (COD or jumps in multiple directions) reproduced in the sports context. In this case, it has been seen that inter-limb asymmetries vary greatly depending on the type of skill. In order to truly measure the asymmetry in a specific way, it should be carried out in the sports context, for example with accelerometers or GPS, technology that is still being developed.

Conclusions

- Please rewrite the whole paragraph. Concentrate on your findings and major implications.

We rewrote the whole paragraph.

“Given the associations between asymmetries and reduced sprint speed and jumping performance, it is suggested that strength and conditioning training interventions should consider the reduction of inter-limb asymmetry in youth team-sport athletes. Moreover, higher asymmetries were associated with diminished jump performance in the weaker leg and appear to be direction-specific. Thus, the fact of having inter-limb asymmetries gives us an opportunity to train and improve from the weaker leg. In addition, practitioners should also consider strengthening the weaker limb during sport-specific skills as well. For example, in the present study basketball athletes are frequently required to perform “lay-ups” which may occur on the weaker limb. Thus, the concept of a window of opportunity for the weaker limb should also be considered within the context of sport skills too.”

---

## [Decision Letter · Decision Letter 1]

18 Nov 2019

PONE-D-19-17934R1

Neuromuscular asymmetries are associated with decrements in physical performance in youth elite team sports athletes

PLOS ONE

Dear Mrs Fort,

Thank you for submitting your manuscript to PLOS ONE. After careful consideration, we feel that it has merit but does not fully meet PLOS ONE’s publication criteria as it currently stands. Therefore, we invite you to submit a revised version of the manuscript that addresses the points raised during the review process.

We would appreciate receiving your revised manuscript by Jan 02 2020 11:59PM. To enhance the reproducibility of your results, we recommend that if applicable you deposit your laboratory protocols in protocols.io, where a protocol can be assigned its own identifier (DOI) such that it can be cited independently in the future. For instructions see: http://journals.plos.org/plosone/s/submission-guidelines#loc-laboratory-protocols

We look forward to receiving your revised manuscript.

Kind regards,

Andreas Mierau, PhD

Academic Editor

PLOS ONE

Reviewers' comments:

Reviewer's Responses to Questions

6. Review Comments to the Author

Reviewer #1: I would like to thank the authors for considering most of my previous comments. However, I still have some concerns. These are detailed below.

Title

Given that you did not directly measure any neuromuscular variables, the word "neuromuscular" seems to not reflect what was really done. Is there any other alternative that could be used here?

Abstract

Line 35 Please add the magnitude of the correlation consistently in the abstract.

Line 36-39 Are these correlation analyses needed as one may assume that asymmetry is associated with the performance of the weaker leg. In other terms, why correlating asymmetry with a factor that was used to calculate asymmetry? It would make sense to examine the association between asymmetry and another measure of physical fitness but with the weaker leg, this is unclear to me.

Line 44 In the discussion you noted " a tendency". Please be consistent with the statements

Introduction

Line 104 Please refer to the exact components of physical fitness used consistently.

Line 110 See my previous comment

Methods

Line 180 Please note that the jump height as a measure of power has been recently questioned in a comprehensive paper published in sports Med

doi: 10.1007/s40279-019-01073-1

You may mention this in the limitation

Line 244 It is rather the magnitude of the difference

Results

Line 285-294 Please refer to my comment in the abstract

Discussion

Line 302-305 This particular statement is hard to grasp. Please rephrase for clarity

Line 305-307 Unclear

310 However instead of despite

313 consider starting a new sentence "This is in accordance....."

319-321 awkward sentence. Please rephrase

321-325 What could be the reason behind this observation? why jump height asymmetry is associated with sprint speed performance but not with CoD performance. This is a comment that I included in my previous report but was not really considered.

329-332 This looks different from what you stated in the previous sentence.

you focused on the association between asymmetry and performance of the weaker leg (which I'm not that convinced why you did it or at least this has to be argued in the rationale) while Bishop et al reported on the correlation between asymmetry and jump performance without mentioning if it was relative to the weaker side or not.

364 "In" instead of at

383-384 Do you mean simply here that the level of asymmetry is more pronounced in females than that in males? if so would be better to express it that way.

404 should be considered with caution instead

Conclusions

424-425 Please reword

Reviewer #2: Dear authors,

thank you for your modifications which improved the quality of your manuscript. Please add the following information to your text body.

- l.186: I do not understand why the OLHT should estimate "dynamic stability". Please

expand.

Dynamic stability is defined as the ability to maintain equilibrium during dynamic

actions (the body is under some kind of displacement), bringing the requirement of

balance to the kind of joint stability involved in sport-specific skills. Here some

references (Noyes et al, 1996; Myer et al, 2011; Munro et al., 2012). We can add to the

text if the reviewer considers it necessary.

- Table 3 and throughout the whole manuscript: I am not sure if the other reviewer really wanted that you use just one decimal for correlations. In my opinion, you should follow this rule of thumb: if you have 2 digits in front of the dot, use 1 decimal; if you have 1 digits in front of the dot, use 2 decimals. Please modify!

Apart from this, everything is fine.

Kind regards

---

## [Author Response · Author response to Decision Letter 1]

9 Dec 2019

Reviewers' comments:

Reviewer #1: I would like to thank the authors for considering most of my previous comments. However, I still have some concerns. These are detailed below.

Title

Given that you did not directly measure any neuromuscular variables, the word "neuromuscular" seems to not reflect what was really done. Is there any other alternative that could be used here?

Ok, we have changed the title and all document to: “Inter-limb asymmetries are associated with decrements in physical performance in youth elite team sports athletes”.

Abstract

Line 35 Please add the magnitude of the correlation consistently in the abstract.

Thanks, we clarified and added the magnitude of the correlation consistently. “Results showed significant (p ˂ 0.05) but small (r = 0.3) relationships between SLCMJ asymmetries and 30 m sprint time for the total group. Significant negative correlations with small to moderate magnitude of correlation were also found between SLCMJ asymmetries and SLCMJ performance on the lowest performing limb for the total group (p < 0.05; r = -0.3), males (p < 0.01; r = -0.5) and females (p < 0.05; r = -0.3). Moreover, significant negative correlations with moderate and large magnitude were also present between OLHT asymmetries and OLHT performance on the lowest performing limb for the total group (p < 0.01; r = -0.4), males (p < 0.01; r = -0.6) and females (p < 0.01; r = -0.6).”

Line 36-39 Are these correlation analyses needed as one may assume that asymmetry is associated with the performance of the weaker leg. In other terms, why correlating asymmetry with a factor that was used to calculate asymmetry? It would make sense to examine the association between asymmetry and another measure of physical fitness but with the weaker leg, this is unclear to me.

We agree with the author. Despite this, this sub-analysis reaffirms the relationship between the weakest leg and a decrease in the physical performance of athletes.

Line 44 In the discussion you noted " a tendency". Please be consistent with the statements

In the discussion we talked about “tendency” since the correlation is significant but the magnitude of this was low in some cases. We removed the adjective "tendency" to not confuse the reader.

Introduction

Line 104 Please refer to the exact components of physical fitness used consistently.

Added: “Therefore, the main objective of this study was to examine the relationship between inter-limb neuromuscular asymmetries and physical performance (Dynamic balance, jumping, linear sprinting, and change of direction speed) in elite youth team-sports players.”

Line 110 See my previous comment.

Added in the previous comment.

Methods

Line 180 Please note that the jump height as a measure of power has been recently questioned in a comprehensive paper published in sports Med

doi: 10.1007/s40279-019-01073-1 

You may mention this in the limitation

We agree with the author. The ability to jump is not the only indicator of the power of the athlete. In this study we have used the jump because it is one of the main motor skills that determine performance in team sports such as volleyball, basketball or handball.

Line 244 It is rather the magnitude of the difference

Changed: “Moreover, the magnitude of the difference in asymmetry between males and females were determined using Cohen’s d effect sizes (ES) using the formula…”

Results

Line 285-294 Please refer to my comment in the abstract

We clarified the text: “Pearson’s correlations (r) between inter-limb asymmetry and test scores are shown in Table 3. Results showed significant (p ˂ 0.05) but small (r = 0.3) relationships between SLCMJ asymmetries and 30 m sprint time for the total group. Significant negative correlations with small to moderate magnitude of correlation were also present between SLCMJ asymmetries and SLCMJ performance on the lowest performing limb for the total group (p < 0.05; r = -0.3), males (p < 0.01; r = -0.5) and females (p < 0.05; r = -0.3). However, the magnitudes of the correlations between SLCMJ asymmetries and SLCMJ performance on the lowest performing limb were small to moderate for the total group (r = -0.3), females (r = -0.3) and males (r = -0.5). Significant negative correlations with moderate and large margnitude were also present between OLHT asymmetries and OLHT performance on the lowest performing limb for the total group (p < 0.01; r = -0.4), males (p < 0.01;r = -0.6) and females (p < 0.01; r = -0.6). In addition, no correlation were observed between SEBT ANT asymmetries and physical performance.” 

Discussion

Line 302-305 This particular statement is hard to grasp. Please rephrase for clarity

We rephrased: “Results showed a tendency that larger jump height asymmetries during the SLCMJ test were associated with slower 30 m sprint times when the entire sample were considered. The same asymmetry index was also associated with reduced jump height on the lowest performing limb for the entire sample and when males and females were separated.”

Line 305-307 Unclear

We clarified: “Larger asymmetries during the OLHT were also associated with reduced distance during the OLHT, also on the lowest performing limb for the whole sample and when separated by males and females.”

310 However instead of despite

Changed.

313 consider starting a new sentence "This is in accordance....."

Changed.

319-321 awkward sentence. Please rephrase

Clarified: “In contrast, in the present study, no significant associations were evident between jump height asymmetry and total time during the V-cut test.”

321-325 What could be the reason behind this observation? why jump height asymmetry is associated with sprint speed performance but not with CoD performance. This is a comment that I included in my previous report but was not really considered.

Clarified: “In contrast, in the present study, no significant associations were evident between jump height asymmetry and total time during the V-cut test. This is also in agreement with previous research. Lockie et al. (16) and Dos’Santos et al. (17) both showed no association between jump height and distance asymmetry from the SLCMJ and single and triple hop tests and COD (505) performance in male collegiate athletes. However, this is in contrast to previous research wich has shown realtionship between major inter-limb asymmetries and COD speed decrements [3,42]. Thus, the lack of consensus regarding the impact of inter-limb asymmetries on COD performance could be explained because of differences in the assessement methods, asymmetry calculations and athlete population.” 

329-332 This looks different from what you stated in the previous sentence.

you focused on the association between asymmetry and performance of the weaker leg (which I'm not that convinced why you did it or at least this has to be argued in the rationale) while Bishop et al reported on the correlation between asymmetry and jump performance without mentioning if it was relative to the weaker side or not.

This is because Bishop et al. presented the data as left vs right (not stronger vs weaker).

364 "In" instead of at

Changed.

383-384 Do you mean simply here that the level of asymmetry is more pronounced in females than that in males? if so would be better to express it that way.

Clarified: “Despite this, our hypotheses was that the level of asymmetry would be more pronounced in females than that in males. However, our hypothesis has been disproven with no significant diffferences between groups in unilateral jumping tasks.”

404 should be considered with caution instead

Changed.

Conclusions

424-425 Please reword

“In addition, practitioners should also consider strengthening the weaker limb during sport-specific skills as well. For example, in a basketball game, performing a “lay-up” on the weaker side (weaker leg) successfully can benefit the performance of the player.”

Reviewer #2: Dear authors, thank you for your modifications which improved the quality of your manuscript. Please add the following information to your text body.

- l.186: I do not understand why the OLHT should estimate "dynamic stability". Please

expand.

Dynamic stability is defined as the ability to maintain equilibrium during dynamic

actions (the body is under some kind of displacement), bringing the requirement of

balance to the kind of joint stability involved in sport-specific skills. Here some

references (Noyes et al, 1996; Myer et al, 2011; Munro et al., 2012). We can add to the

text if the reviewer considers it necessary.

The stability component of the OLHT test is not in the jump capacity but in the stability that the lower limb should provide just after a maximum unilateral jump (subjects must hold the balance for two seconds after landing).

- Table 3 and throughout the whole manuscript: I am not sure if the other reviewer really wanted that you use just one decimal for correlations. In my opinion, you should follow this rule of thumb: if you have 2 digits in front of the dot, use 1 decimal; if you have 1 digits in front of the dot, use 2 decimals. Please modify!

We are totally agreed. But this is the comment of reviewer 1. “Would be better to round the values to one single decimal after the dot. Please, do this consistently throughout the manuscript.”. If the reviewer consider we have to change it is not a problem for us.

---

## [Decision Letter · Decision Letter 2]

7 Jan 2020

PONE-D-19-17934R2

Inter-limb asymmetries are associated with decrements in physical performance in youth elite team sports athletes

PLOS ONE

Dear Mrs Fort,

Thank you for submitting your manuscript to PLOS ONE. After careful consideration, we feel that it has merit but does not fully meet PLOS ONE’s publication criteria as it currently stands. Therefore, we invite you to submit a revised version of the manuscript that addresses the points raised during the review process.

We would appreciate receiving your revised manuscript by Feb 21 2020 11:59PM. To enhance the reproducibility of your results, we recommend that if applicable you deposit your laboratory protocols in protocols.io, where a protocol can be assigned its own identifier (DOI) such that it can be cited independently in the future. For instructions see: http://journals.plos.org/plosone/s/submission-guidelines#loc-laboratory-protocols

We look forward to receiving your revised manuscript.

Kind regards,

Andreas Mierau, PhD

Academic Editor

PLOS ONE

Journal Requirements:

Additional Editor Comments (if provided):

As for the decimals, please follow the recommendation of reviewer 2 (i.e. if you have 2 digits in front of the dot, use 1 decimal; if you have 1 digit in front of the dot, use 2 decimals)

Reviewers' comments:

Reviewer #1: I thank the authors for responding to my comments. After reading the revision, I noted some further comments and suggestions.

Major concern

I have raised the concern about correlating the asymmetry with the performance of the weaker leg in my previous review report. I find authors’ answer not convincing in this context. I still think that the correlation analyses authors’ have made between asymmetry and the outcome of weaker side do not have a real significance. Above all, the weaker leg was used to determine asymmetry. Then, a significant correlation is not surprising.

Methods

Considering the criticisms related to the ability of jump height to predict muscle power, I would recommend to mention this point in the limitations of the study.

Discussion

Line 309: instead of tendency which was already removed, I suggest to note the range of magnitude (e.g., small-to-moderate)

Line 310: rather “lower 30 m sprint performance”. It may not be that clear to talk about “slower time”.

Line 333 spelling error

In a previous reply authors noted : “ this is because Bishop et al presented the data as left vs right (not stronger vs weaker)”. Sorry, may be I missed something, but I still do not get what you want to express. So, Bishop et al introduced the data as left vs right, then how can the reader knows the weaker from the stronger side?

Reviewer #2: - l.186: I do not understand why the OLHT should estimate "dynamic stability". Please

expand.

Dynamic stability is defined as the ability to maintain equilibrium during dynamic

actions (the body is under some kind of displacement), bringing the requirement of

balance to the kind of joint stability involved in sport-specific skills. Here some

references (Noyes et al, 1996; Myer et al, 2011; Munro et al., 2012). We can add to the

text if the reviewer considers it necessary.

The stability component of the OLHT test is not in the jump capacity but in the stability that the lower limb should provide just after a maximum unilateral jump (subjects must hold the balance for two seconds after landing).

 As indicated in line 202 and Table 2, you analysed the greatest distance for OLHT. This is in my opinion no measure for dynamic stability. Please expand why you analysed the jump capacity, although you emphasize the stability component of OLHT. Please rephrase l.195! What happened if participants performed 2 invalid jumps with one leg?

- l. 206: Please state where the front foot was placed in relation to the first timing gate.

- Table 3 and throughout the whole manuscript: I am not sure if the other reviewer really wanted that you use just one decimal for correlations. In my opinion, you should follow this rule of thumb: if you have 2 digits in front of the dot, use 1 decimal; if you have 1 digits in front of the dot, use 2 decimals. Please modify!

We are totally agreed. But this is the comment of reviewer 1. “Would be better to round the values to one single decimal after the dot. Please, do this consistently throughout the manuscript.”. If the reviewer consider we have to change it is not a problem for us.

 Please contact the editor to ask for journal guidelines concerning this issue.

---

## [Author Response · Author response to Decision Letter 2]

13 Jan 2020

Additional Editor Comments (if provided):

As for the decimals, please follow the recommendation of reviewer 2 (i.e. if you have 2 digits in front of the dot, use 1 decimal; if you have 1 digit in front of the dot, use 2 decimals)

Changed.

Reviewers' comments:

Reviewer #1: I thank the authors for responding to my comments. After reading the revision, I noted some further comments and suggestions.

Major concern

I have raised the concern about correlating the asymmetry with the performance of the weaker leg in my previous review report. I find authors’ answer not convincing in this context. I still think that the correlation analyses authors’ have made between asymmetry and the outcome of weaker side do not have a real significance. Above all, the weaker leg was used to determine asymmetry. Then, a significant correlation is not surprising.

We agree with the reviewer and have added the following information to the limitations: 

“Another finding of this study is that larger jump height asymmetries (vertical and horizontal directions) are associated with diminished jump performance; specifically in the weaker leg and appear to be direction-specific. However, these results do have some limitations that practitioners should be aware of. The significant correlation between asymmetry and the weaker leg is to be expected, given that asymmetry is ratio of both the stronger and weaker limbs. Therefore, it is suggested that correlations with ratios should be made with independent variables (e.g., jump height asymmetry vs. speed or CODS). Despite this, these results are in accordance with Bishop et al. (7) who also showed that larger jump height and distance asymmetries (from the SLCMJ and triple hop tests) were also associated with reduced jump performance during the SLCMJ and horizontal hop tests respectively [7]. In this case, they found decrements in performance in both limbs, which is contrary to our study, where significant correlations were noted in the weaker limb only. The differences in results between studies could be explained because Bishop et al. (7) presented the data as left vs. right (not stronger vs. weaker). 

Methods

Considering the criticisms related to the ability of jump height to predict muscle power, I would recommend to mention this point in the limitations of the study.

Thanks for the consideration, but measure muscle power was not the aim of our study. We modified this sentence: “The one leg hop for distance is used to estimate unilateral jump capacity.”

Discussion

Line 309: instead of tendency which was already removed, I suggest to note the range of magnitude (e.g., small-to-moderate)

Added. “Results showed that larger jump height asymmetries during the SLCMJ test were associated (small-to-moderate magnitude) with lower 30 m sprint performance when the entire sample were considered.”

Line 310: rather “lower 30 m sprint performance”. It may not be that clear to talk about “slower time”.

Changed before. 

Line 333 spelling error

In a previous reply authors noted : “ this is because Bishop et al presented the data as left vs right (not stronger vs weaker)”. Sorry, may be I missed something, but I still do not get what you want to express. So, Bishop et al introduced the data as left vs right, then how can the reader knows the weaker from the stronger side?

Please see our amendments above re: your additional major concern where hopefully this now has been addressed. Thank you for considering our stance on this.

Reviewer #2: - l.186: I do not understand why the OLHT should estimate "dynamic stability". Please expand.

Dynamic stability is defined as the ability to maintain equilibrium during dynamic

actions (the body is under some kind of displacement), bringing the requirement of

balance to the kind of joint stability involved in sport-specific skills. Here some

references (Noyes et al, 1996; Myer et al, 2011; Munro et al., 2012). We can add to the

text if the reviewer considers it necessary.

The stability component of the OLHT test is not in the jump capacity but in the stability that the lower limb should provide just after a maximum unilateral jump (subjects must hold the balance for two seconds after landing).

 As indicated in line 202 and Table 2, you analysed the greatest distance for OLHT. This is in my opinion no measure for dynamic stability. Please expand why you analysed the jump capacity, although you emphasize the stability component of OLHT. Please rephrase l.195! What happened if participants performed 2 invalid jumps with one leg?

We agree with the author, we delete the component of dynamic stability in the text, this is not the aim of the test. “The one leg hop for distance is used to estimate unilateral jump capacity.” We also added. “If participants performed 2 invalid jumps with one leg, they had the opportunity to jump as many times as necessary until they kept their balance.”

- l. 206: Please state where the front foot was placed in relation to the first timing gate.

Added. “The front foot was placed 0.5 m before the first timing gate.” 

- Table 3 and throughout the whole manuscript: I am not sure if the other reviewer really wanted that you use just one decimal for correlations. In my opinion, you should follow this rule of thumb: if you have 2 digits in front of the dot, use 1 decimal; if you have 1 digits in front of the dot, use 2 decimals. Please modify!

We are totally agreed. But this is the comment of reviewer 1. “Would be better to round the values to one single decimal after the dot. Please, do this consistently throughout the manuscript.”. If the reviewer consider we have to change it is not a problem for us.

 Please contact the editor to ask for journal guidelines concerning this issue.

Thanks. Changed in all document.

---

## [Decision Letter · Decision Letter 3]

7 Feb 2020

Inter-limb asymmetries are associated with decrements in physical performance in youth elite team sports athletes

PONE-D-19-17934R3

Dear Dr. Fort,

We are pleased to inform you that your manuscript has been judged scientifically suitable for publication and will be formally accepted for publication once it complies with all outstanding technical requirements.

With kind regards,

Andreas Mierau, PhD

Academic Editor

PLOS ONE

---

## [Editor Report · Acceptance letter]

20 Feb 2020

PONE-D-19-17934R3 

Inter-limb asymmetries are associated with decrements in physical performance in youth elite team sports athletes 

Dear Dr. Fort-Vanmeerhaeghe:

I am pleased to inform you that your manuscript has been deemed suitable for publication in PLOS ONE. Congratulations! Your manuscript is now with our production department. 

With kind regards,

on behalf of

Dr. Andreas Mierau 

Academic Editor

PLOS ONE